# Irruption Theory: A Novel Conceptualization of the Enactive Account of Motivated Activity

**DOI:** 10.3390/e25050748

**Published:** 2023-05-02

**Authors:** Tom Froese

**Affiliations:** Embodied Cognitive Science Unit, Okinawa Institute of Science and Technology Graduate University, 1919-1 Tancha, Onna-son 904-0495, Okinawa, Japan; tom.froese@oist.jp

**Keywords:** autopoiesis, adaptivity, embodied cognition, motivated activity, entropy

## Abstract

Cognitive science is lacking conceptual tools to describe how an agent’s motivations, as such, can play a role in the generation of its behavior. The enactive approach has made progress by developing a relaxed naturalism, and by placing normativity at the core of life and mind; all cognitive activity is a kind of motivated activity. It has rejected representational architectures, especially their reification of the role of normativity into localized “value” functions, in favor of accounts that appeal to system-level properties of the organism. However, these accounts push the problem of reification to a higher level of description, given that the efficacy of agent-level normativity is completely identified with the efficacy of non-normative system-level activity, while assuming operational equivalency. To allow normativity to have its own efficacy, a new kind of nonreductive theory is proposed: irruption theory. The concept of irruption is introduced to indirectly operationalize an agent’s motivated involvement in its activity, specifically in terms of a corresponding underdetermination of its states by their material basis. This implies that irruptions are associated with increased unpredictability of (neuro)physiological activity, and they should, hence, be quantifiable in terms of information-theoretic entropy. Accordingly, evidence that action, cognition, and consciousness are linked to higher levels of neural entropy can be interpreted as indicating higher levels of motivated agential involvement. Counterintuitively, irruptions do not stand in contrast to adaptive behavior. Rather, as indicated by artificial life models of complex adaptive systems, bursts of arbitrary changes in neural activity can facilitate the self-organization of adaptivity. Irruption theory therefore, makes it intelligible how an agent’s motivations, as such, can make effective differences to their behavior, without requiring the agent to be able to directly control their body’s neurophysiological processes.

## 1. Introduction

What is action? Intuitively, an action is a movement that is motivated, e.g., it is completed by an agent for a reason, in contrast to a bodily movement that the agent merely undergoes without there being any motivation for it. In Kauffman’s [1] memorable phrasing of this distinction, an action is an agent’s “doing”, rather than merely a physical “happening”. This distinction brings to the foreground questions about normativity and its biological basis. In contrast to the happenings that are described by physics, which are simply events, an action is also a kind of motivated activity. This implies that there are normative criteria with respect to how well the movements involved in an action are satisfying its motivations, such as conditions of accuracy, success, communicative intent, beauty, or any other motivation.

As an illustrative contrastive example, consider that it makes no sense to attribute a failure to a river if it happened to run out of water; yet it would make sense to attribute failure to an animal’s actions if it was unable to satisfy its need of water. Arguably, in the latter case, it is even a case of intrinsic rather than just externally attributed failure; that there is something seriously going wrong would be manifested by impaired functioning of the body, and this in turn would be given to the animal’s own concerned perspective on the situation. Lastly, it is important to keep in mind that we also have our own first-person perspective on the difference between bodily doings and happenings; while our lived experience reveals that both kinds of bodily activity can be associated with a sense of ownership (i.e., this is my body that is moving), only the former is also associated with a sense of agency (i.e., I am moving my body). Accordingly, our conscious doings are a complex kind of motivated activity.

In general terms, therefore, we can define motivated activity as any embodied activity that is directed at an end, including any activity that the agent performs because of a goal, belief, desire, feeling, thought, experience, value, or any other reason. Although this is not our current focus, it naturally also includes all the regulatory activity of the organism. It can be operationally characterized in general terms as follows: “Intuitively, motivated activity is a particular kind of end-directed activity that is driven by the intrinsic preferences of the system in question” [2] (p. 90). On this general view, we can also consider all cognitive activity studied by cognitive science as a kind of motivated activity.

How do we account for this normative dimension of motivated activity compared to mere happenings? Moreover, how can motivations make a difference to behavior? These questions become especially pressing in the case of ourselves—conscious adult humans—because we also have a first-person perspective on our own motivated activity, which is related to our sense of agency. Our lived experience, therefore, adds to the challenge of how to account for motivated activity in such a way that our lived motivations, as such, can be understood as participating in the generation of the behavior. We leave the development of an account of our experiential dimension of motivated activity as a topic for future work; here, the focus is on the efficacy of motivated activity. Below, we review the enactive approach to motivated activity and its criticisms, and then propose a modified account by introducing the concept of irruption to capture the role of motivations.

## 2. The Enactive Approach to Motivated Activity

As part of its overall ambition to bring human experience closer to cognitive science [3], the enactive approach has been developing an influential account of motivated activity for over two decades now. It started with a revitalization of autopoietic theory as a biology of intentionality [4], and then continued with the introduction of normative concepts from the philosophy of the organism, especially Kant’s concept of natural purposes and Jonas’s [5] notions of precariousness and needful freedom [6]. A common theme of the enactive approach is the autonomy of the living system. This biological autonomy is associated with the fact that the living system’s systemic organization, or form, is actively maintained across time, while the matter instantiating the form is continuously replaced. Importantly, this process of systemic and material self-production is understood as a minimal kind of motivated activity:

“This autonomy then is nothing other than true teleological behavior. This autonomy has to do with the ever-existing gap between the realization of the living and its underlying matter. […] there is always the possibility, and final certainty, of death. It is this existential situation that is emphasized by Jonas: the teleological, circular, self-referential movement of the living. To live means to say yes to oneself emphatically as the basic movement of existence, because existence is always existence of form on and against pure matter.” [6] (p. 119).

This poetic description of the living condition was a key source of inspiration for the tradition of theorizing that has led to the contemporary enactive conception of life [7]. In short, it is because an organism metabolically brings about its own individual existence, under conditions of precariousness that require continuous regulation of its internal and behavioral activity, that features an intrinsic perspective to which things matter in the first place. In this way, autopoiesis and adaptivity ground the organism’s intrinsically value-laden perspective on the world, which is referred to as a process of sense-making. Accordingly, Di Paolo, Rohde, and De Jaegher [8] proposed to define value as “the extent to which a situation affects the viability of a self-sustaining and precarious process that generates an identity”.

Much work on the enactive approach to value has focused on how to account for its presence, while much less attention has been paid to how, once present, value could make a concrete difference to an organism’s activity. The new theory of motivated activity to be developed in this article focuses on this latter topic, i.e., on efficacy, while taking the presence of motivations, such as value, for granted. This has the advantage that the arguments do not depend on any specific account of the origins of motivations, which even opens the possibility that the arguments could be deployed more widely than for the enactive approach that forms the current starting point. Thus, how should the role of this value in the organism’s concerned perspective be conceptualized? As Di Paolo and colleagues highlighted, in contrast to standard approaches to cognitive neuroscience that tend to reify this role as a value system in the brain with localized functions, the enactive approach proposes a holistic system-level role that shapes all the organism’s sense-making activities (Figure 1).

One subsequent conceptual development is noteworthy here. In its initial formulations, life’s precariousness was attributed in an unspecific way to “the ever-existing gap between the realization of the living and its underlying matter” [6] (p. 119). This idea was subsequently refined in the enactive conception of life in terms of a process of self-individuation [7,9]; there is always already a primordial tension at the core of life between two opposite needs with respect to how an organism must relate to its environment (openness versus closedness; see Figure 2 for a schematic diagram). This tension introduces an additional normative constraint, namely, for the living system to have the capacity to adaptively coordinate the partial satisfaction of these two opposing needs over time, which can also be considered as a minimal form of agency. The adaptivity needed to achieve this has been taken to involve a process of active monitoring, possibly based on a dedicated mechanism, which allows for the discrimination of system tendencies in terms of how threatening they are to the maintenance of its identity, and to accordingly compensate for deleterious tendencies [9,10].

In addition, the enactive approach has long emphasized that life is a process of open-ended becoming across all scales [12], and this historicity is realized by path-dependent plasticity of a living system’s parameters, variables, and constraints [13]. In order to highlight the role of this plasticity for the enactive conception of life, Froese, Weber, Shpurov, and Ikegami [11] redrew Figure 2, originally by Di Paolo et al. [9], by including plasticity as an explicit aspect of the way in which agency resolves the primordial tension via coordination of partial constraint satisfaction over time (Figure 3).

In summary, given that autopoietic theory was originally formulated in explicitly antiteleological terms in the context of Maturana’s biology of cognition [14], this “normative turn” of the enactive approach amounts to a major departure from that tradition [15]. The enactive approach continues to share its commitment to a rejection of representationalist and other homuncular style explanations in biology, but it has distanced itself from an account that reduces all of an organism’s activity to nothing but the happenings of structural coupling shaped by neutral drift. To make room for a genuine role of agency, e.g., as is it given to us in our first-person perspective on motivated activity, the enactive approach has opted to create conceptual space that allows for an emergent role of normativity at the system level of the whole organism. On one side, this normative turn has given rise to a vibrant enactive research community, which has continued to develop the implications of this conception of life, especially by developing its extension into the domain of social interaction [16,17]. On the other hand, this normative turn has also given rise to a series of concerns which highlight that there continues to be a demand for refinement of its theoretical foundations. Here, we focus on the specific enactive approach to biological intentionality sometimes known as autopoietic enactivism. Radical enactivism aims to naturalize an organism’s intentional directedness in terms of a history of selection [18], but this move raises its own set of concerns (e.g., [19]).

## 3. Concerns about the Normative Turn of the Enactive Approach

It is interesting that the gist of these concerns pulls into two opposite directions. On the one hand, there is a general worry that the enactive concept of normativity falls short of being able to accommodate the full richness of life, mind, and especially human experience. For instance, the current account of normativity arguably remains too closely tied to the notion of self-production of an identity, which makes it challenging, for example, to account for life’s tendency to go beyond itself [15], intentionality [20], and concern for others [21]. To be fair, the primordial tension of the metabolic core of the organism can become re-expressed in the form of dialectical dynamics at higher levels of self-individuation, including in sensorimotor and social modes of life [9,17]. However, what remains without a deeper explanation is what motivates the generation of behaviors that eventually give rise to these new identities of the organism in the first place. Something other than just self-production and homeostasis seems to be required to explain these transitions in individuality [22].

This tendency toward open-ended becoming and progressive self-individuation seems to call for an extension of the enactive approach’s naturalist conceptual toolkit, e.g., with a maximizing principle akin to that of maximum entropy production [2], or even simply the second law of thermodynamics, which has been deployed in other contexts to explain the tendency of life and mind to increase in complexity [23]. Yet, compared to other related frameworks, such as ecological psychology [24] and interactionism [25], there has been notably little interest by proponents of the enactive approach to engage more deeply with these advances in the thermodynamics of life. This lack of interest has deep roots, with Maturana and Varela [14] explicitly distancing autopoietic theory from its material basis [26], while contemporary frameworks promoted a close integration of dynamics and thermodynamics [27]. While the enactive conception of life has benefited from a closer engagement with the physics of life, there seems to be little urgency in further unpacking the details. Perhaps this reticence is not unrelated to the long-standing debate about the extent to which the enactive approach is even committed to naturalism, especially given that its close alliance with phenomenological philosophy seems to bring with it certain idealist [28] or anti-naturalist [29] tendencies. Certainly, some would prefer the enactive approach to keep even more distance from the naturalist–objectivist framework of science [30] or, alternatively, to reconceive the very concept of nature and, hence, to opt for a more relaxed kind of naturalism that allows phenomena pertaining to mind and sociality to also be counted as part of nature [31,32].

On the other hand, it is precisely these kinds of considerations which can raise the worry that the enactive approach goes too far in trying to accommodate human experience in cognitive science. Some enactive terminology, such as the popular concept of sense-making, is not far from explicitly representationally loaded concepts that are used in philosophy of mind [18]. Furthermore, this does not seem to be merely a terminological issue; the way in which adaptivity was originally defined, i.e., in terms of processes of active monitoring and discrimination of tendencies regarding their impact on viability, makes it difficult to discern how the enactive approach differs from the kind of representational approach it supposedly rejects [33]. It does not help that the general notion of sense-making, which motivated the elaboration of autopoietic theory in the direction of adaptivity [10], was in turn motivated by the richly graded normativity of human lived experience. An appeal to the characteristics of our first-person perspective can make it harder to see the generality of the scientific concepts, or at least it would require further theoretical work to demonstrate the generality. Instead, it appeals to human lived experience to motivate fundamental concepts in biology that are supposed to be characteristic of life as such, opening the enactive approach to the serious worry that its conception of life is anthropomorphically biased and, hence, will ultimately turn out to be scientifically unworkable [34].

The enactive approach to life, therefore, finds itself in the seemingly impossible position that its critics are pulling into two opposing directions: (1) to add more psychological relevance to its biological foundations, even if this would mean foregoing its commitment to a strict kind of naturalism, and (2) to remove all psychological inspiration from its biological foundations, such that there is no longer a problem of naturalization in the first place. However, despite these criticisms, most proponents of the enactive approach do not intend (1) to turn it to another form of cognitivism, vitalism, or even panpsychism, or (2) to return to the biology of cognition or even a form of physicalism. Yet, the middle way that the enactive approach has been developing does not yet have a systematic response to these criticisms either.

Consider the enactive concept of value by Di Paolo and colleagues [8]; they reject the reification of value into a local function of appraisal that generates input to other processes, in favor of a role of value as a global constraint on all processes. However, when developing this alternative framework, the general theoretical move of identifying value with an aspect of the material living system is retained (see Figure 1 where “value” is tellingly written inside both panels A and B, illustrating the cognitivist and enactive frameworks of action, respectively). The main difference is only whether the role of value is identified with a local or a global property. Responding to the two sets of criticisms could then go in two directions; it could either involve making the role of value more prominent by spelling out in more detail how a global value makes an operational difference (e.g., by elaborating a suitable cognitivist architecture, such as global workspace theory) or entail removing the appeal to a role of value from inside the picture altogether (e.g., by replacing it with a nonteleological organizing principle of nature, such as maximum entropy production).

The enactive approach has not pursued either of these possible responses, but not for a lack of opportunities. There are other frameworks that are explicitly sympathetic to the enactive approach, but that are also more open to forms of representation [25,35], or that would connect its ideas more directly with considerations drawn from physics [2,36]. Why does the enactive approach resist either of these directions, even if it means leaving both sets of criticisms unanswered?

## 4. Toward a Motivation-Involving Account of Motivated Activity

One possible diagnosis of the enactive approach’s unwillingness or incapacity to respond to these critics by pursuing one of the existing alternatives is that this would be in tension with another distinctive feature of the enactive approach, namely, its insistence on allowing for the efficacy of the subjective dimension of life. For example, in Di Paolo, Rohde, and De Jaegher’s [8] assessment, human lived experience is identified as one of the five cores idea of the enactive approach: “Far from being an epiphenomenon or a puzzle as it is for cognitivism, experience in the enactive approach is intertwined with being alive and enacting a world of significance”. This commitment to the role of lived experience has been made most explicit in the context of its neurophenomenology research program:

“A distinctive feature of this neurophenomenological approach is that it allows conscious activity to be a causally efficacious participant in the cycles of operation constitutive of the subject’s life.” [37].

More recently, Fuchs [38,39] has worked extensively on this unique ambition of the enactive approach by developing a framework of dual aspectivity that aims to integrate human experience and biological embodiment. Yet, at this point, it is still not even clear how we should conceive of the possibility that an agent’s motivations, as such, are efficacious in its material embodiment. Nevertheless, the very fact that the enactive approach explicitly aims at this radical possibility makes all the difference. In the context of the cognitive science of motivated activity, we can call this envisioned possibility a “motivation-involving” account. The ambition to develop such a motivation-involving account entails important constraints regarding what would count as an acceptable account. Specifically, this explanatory ambition to account for the involvement of motivation, as such, in an agent’s activity makes it intelligible why the enactive approach cannot fully commit either (1) to identify the efficacy of the first-person perspective, or that of agency more generally, with the efficacy of postulated sub-personal representational processes (e.g., cognitivism), or (2) to identify it directly with the efficacy of neurobiological processes (e.g., biology of cognition). Both existing explanatory strategies would ultimately undermine the possibility for an agent’s motivation itself to have a causally efficacious involvement in a behavior’s material processes.

This diagnosis of the search for a “motivation-involving” account would also help to explain the enactive approach’s uneasy relationship with the field of artificial life. Simulation models have always been welcome tools for undermining the necessity claims of representational explanations of motivated activity, i.e., by implementing a functionally equivalent, nonrepresentational alternative in terms of dynamical systems [40]. In particular, in its initial phases, the enactive approach was keen on devising simulated “thought experiments” [41], which can serve as proof of concept that internal mental representations are not necessary to generate a behavior of interest. Work in artificial life has also positively served to sharpen key enactive concepts such as autonomy [42], habits [43], and sensorimotor contingencies [44].

Yet, there also seems to be an unacknowledged danger that these antirepresentational simulation models can go too far in the other extreme, e.g., by showing that adaptivity can be implemented in computer models of artificial chemistry [45] and cellular automata [46] without any need to appeal to notions of active monitoring or discrimination of tendencies related to viability. The general upshot of this kind of work is that “dynamics alone is sufficient for adaptivity, and no explicit adaptive process is required” [46] (p. 205), which is a slippery slope from the enactive conception of life back to the autopoietic theory of Maturana’s biology of cognition from which it had explicitly distanced itself. Admirable attempts at promoting a more enactive interpretation of this kind of artificial life modeling by dressing up system diagrams with explicitly normative labels such as “value” [8] and “adaptive region” [47] ultimately do not make room for anything but the dynamics of mere happenings [48]. We can, therefore, understand why some proponents in the enactive approach became interested in putting the human back in the loop of their research by switching focus from artificial life to human–computer interfaces [49].

In summary, competing explanatory frameworks in the sciences of life and mind, including cognitivism and biology of cognition, rule out motivation-involving accounts of motivated activity, each in their own way by basically offloading the efficacy of motivation to that of internal system components and processes. The enactive approach has, therefore, responded by attempting to make conceptual space for the causally efficacious involvement of an agent’s motivations at the global system level of the organism embodied in its world. This is best seen again in the context of the neurophenomenology research program:

“Given that the coupled dynamics of brain, body, and environment exhibit self-organization and emergent processes at multiple levels, and that emergence involves both upward and downward causation, it seems legitimate to conjecture that downward causation occurs at multiple levels in these systems, including that of conscious cognitive acts in relation to local neural activity.” [50] (p. 421).

Thompson and Varela go on to describe consciousness as an emergent phenomenon, akin to an order parameter that mediates relations among neurons. What remains unclear, however, is where we should attribute the causally efficacious “effects of a moment of consciousness and its substrate large-scale neural assemblies” [50] (p. 421). More importantly, it ultimately does not seem to matter whether we assign the subject’s efficacy to a collective property (order parameter) or to the efficacy of a material property (neural substrate). Either way, there is simply no conceptual room for a motivation-involving account if all efficacy is offloaded to a nonmotivational organizing property of the material body, while operational equivalence is assumed. So far, neurophenomenology has not been able to offer an account of how a subject’s consciousness could make a difference in its own right to the activity of their brain and body [51].

Not surprisingly, therefore, the theoretical problems faced by a motivation-involving account do not disappear by replacing bottom-up causation with top-down causation. This move may have made some conceptual progress by at least allowing for the possibility that factors other than local causal interactions play a role, but it does so by adopting the contentious concept of downward causation and without resolving the core problem. In other words, additional work would be required to demonstrate that something like an order parameter can do causal work of its own, rather than simply serving as a usefully concise redescription of local dynamics. Worse, even by accepting this additional explanatory burden, the proposal ends up pushing the explanatory problem of the efficacy of motivation to a higher level of description; the question would then become how the motivation, as such, makes a difference to the system at that collective level, e.g., by somehow modulating the putative causal workings of the order parameter. Yet, there is currently no compelling explanation of how an order parameter could serve this role [52].

The enactive approach is not alone with these theoretical issues; in this respect, it overlaps with a class of research programs that, following Thompson [53], we can call embodied dynamicism. A key feature of these theoretical frameworks is that they are characterized by an appeal to emergent top-down constraints in nonlinear dynamical systems as a way of explaining motivated activity (e.g., [54,55,56,57,58,59]). Yet, at the same time, the enactive approach stands apart because of its commitment to a special kind of non-reductionism, i.e., what we have been calling a motivation-involving account of behavior, which existing frameworks cannot provide.

Therefore, to properly secure a motivation-involving account, it seems necessary to go back to the theoretical drawing board and start again from first principles, such that motivational involvement is not already ruled out by preexisting commitments before it can get properly formulated. This move will not fit everyone’s inclination, as some proponents of the enactive approach prefer to settle for a kind of mind/brain identity theory that is phenomenologically enriched [60], embodied [61], and situated [62]. However, there are also those who are feeling more adventurous and aim to make room for the possibility that our motivations play a distinctive role in shaping behavior, especially by developing enactive ideas in the direction of a libertarian philosophy of human freedom and agency (e.g., [39,52,63]).

Inspired by these latter authors, I propose to open a new branch in the tree of enactivisms dedicated to the search for a motivation-involving account of motivated activity in the context of a relaxed naturalism that allows both an agent’s motivations and its materiality to be part of reality. This account needs to face the classic libertarian double challenge: how to reject the complete material determination of agent behavior, while avoiding falling into the opposite extreme of completely random agent behavior. Either of these options of the classic dilemma of the libertarian philosophy of free will would rule out the efficacy of motivations for behavior. The remainder of this article introduces the foundations of a novel systematic framework that solves this double challenge: “irruption theory”.

## 5. Irruption Theory

As a useful starting point, we can build on Fuchs’ [39] analyses conducted in his book *In Defense of the Human Being*. A core insight of his defense is that “the freedom to decide and act is a primary experience of our everyday life” [39] (p. 124). In turn, it is accepted that, in general, our lived experience matters to us; it is in virtue of how we consciously experience things that we are motivated to behave in certain ways and decide to act one way or another, a position that Cleeremans and Tallon-Baudry [64] call “phenomenal efficacy”. We can agree with Cleeremans and Tallon-Baudry on the point of phenomenal efficacy, but they instead opt for a functionalist and compatibilist explanation, possibly to avoid this tension. However, given that they identify conscious mental states with global, high-level states of an organism that are assumed to have their own causal efficacy, it can be expected that their account will eventually run into the kinds of explanatory issues faced by embodied dynamicism, as discussed in the previous section.

Thus, in apparent tension with this phenomenal efficacy, no matter how closely we investigate the body of a conscious person, their neurophysiological processes “inevitably remain external or opaque for us—nowhere does something like subjective experience emerge” [39] (p. 21). However, this recognition that we can only ever measure a person’s material features does not entail that all behavior must, therefore, be fully determined by physical laws, nor that calculations based on those measures could make behavior fully predictable.

The level of human deliberations, reasons, motives, and intentions simply does not appear in the calculations; likewise, their effectiveness remains *physically unobservable*. Therefore, in order to secure our experience of freedom, we only need to assume that the neuronal carrier processes are *not exclusively* determined by physical laws or, in other words, that actions are *physically underdetermined* [39] (p. 139).

With these preliminary remarks, we arrive at a basic libertarian position. The task for irruption theory is to unpack Fuchs’ assumptions and to take some additional steps that go beyond securing the mere possibility of a motivation-involving account. What is needed to strengthen this position is to show how it can be developed it into a scientifically productive account of motivated activity, which in turn depends on making intelligible the role played by the agent’s motivations in the material underdetermination of behavior such that the agent’s behavior is both motivated and effective.

### 5.1. Axioms

The first two axioms of irruption theory are as follows:

Axiom 1: Motivational efficacy. An agent’s motivations, as such, make a difference to the material basis of the agent’s behavior.

Axiom 2: Incomplete materiality. It is impossible to measure how motivations, *as such*, make a difference to the material basis of behavior.

Each of these axioms seems to be independently valid; Axiom 1 derives most of its support from our first-person perspective on motivated activity, while Axiom 2 derives most of its support from a third-person observation of motivated activity. Taken together, they seem to be in tension because Axiom 2 does not offer direct empirical support for Axiom 1. Accordingly, motivated activity is an explanatory challenge whose solution seems to require explicitly or implicitly undermining the validity of these axioms. Strategies include undermining Axiom 1 by turning the lived experience into an illusion or fiction, weakening Axiom 1 by identifying the efficacy of the motivation with that of a material surrogate (global or local property), or weakening both axioms by postulating internal mental representations in the brain as the computational implementation of the motivation. These strategies are all beset by theoretical problems that have not yet yielded a satisfactory solution, and this is not the place to review them (see [18,65]). Importantly, however, given that the two axioms are independently compelling, it would be preferable to find a way to dissolve their tension without undermining either of them.

Irruption theory takes both axioms as valid, but it demarcates the scope of both axioms such that the tension is diffused. Specifically, Axiom 1 would be in tension with Axiom 2 only if we could experience precisely how our motivational involvement makes a difference to bodily activity; yet, this is not the case [66]. When we lift our finger, it simply rises; we cannot experience our action at the level of its neurophysiology, no matter how much we bring preconscious processes into awareness. More importantly, Axiom 2 is in tension with Axiom 1 only with the added assumption of determinism, i.e., of a complete causal closure of material processes such that motivational involvement in behavior is already ruled out in principle. While determinism is often taken for granted in cognitive science, it is not universally accepted. For example, there are compelling arguments that the notion of a clockwork universe has been invalidated by the most successful natural science, quantum physics [67]. In line with libertarian positions, irruption theory takes an explicit stance in this debate by adopting a third axiom:

Axiom 3: Underdetermined materiality. An agent’s behavior is underdetermined by its material basis.

The precise extent of the material underdetermination of an agent’s behavior remains to be established, but there will crucially be an irreducible limit to predicting its next state on the basis of current material conditions alone. With Axiom 3 in place, the apparent tension between Axioms 1 and 2 has lost some of its force: Axiom 2 can no longer be interpreted in the sense of evidence of absence counting against Axiom 1, given that the material basis itself could never amount to a complete determination of behavior either. If we were to stop at this conclusion, then we would have at least gained a truce of logical compatibility between phenomenology and cognitive science [39].

Irruption theory goes further by proposing to make the behavior’s underdetermination the target of a dedicated research program. By shifting cognitive science from its focus on Axiom 2 to a systematic engagement with Axiom 3, the ambition is to uncover hidden degrees of freedom in the agent’s material basis that are associated with motivational involvement, as per Axiom 1, but without coming into tension with Axiom 2. What Axiom 3 implies is that, in superficial analogy to the quantum revolution in physics, we need to start working with unpredictability as inherent to the motivated activity of life and mind, rather than explain it away as deriving from insufficient experimental control or inaccurate measurement. In this sense, irruption theory is a radical response to the need for more “big theory” to address the replication crisis in psychology at a fundamental level [68].

### 5.2. Theses

The innovative contribution of irruption theory is to work with the relative uncertainty of motivated activity. In line with Axiom 3, it is widely accepted that there are background levels of fluctuations in all material processes, including in those underlying motivated activity. The key idea is then the following: if involvement of motivations makes a difference to an agent’s embodied activity in its own right, i.e., in a way that is not reducible to its material basis, then this should be associated with higher levels of underdetermination of that activity by its material basis. This key idea leads to the following working hypothesis:

**Working Hypothesis of Irruption Theory:** 
*The more an agent’s embodied activity is motivated, the less that activity is determined by its material basis.*


To capture this complementary relationship between the motivations and materiality of an agent’s activity, a new concept is introduced: *irruption*. The word “irruption” was chosen because the original Latin meaning refers to a violent or sudden bursting or breaking into, which recalls the “breaks” famously introduced by Ashby [69] to account for adaptivity (more on breaks below, in the context of attunement). This meaning also echoes the “cross-border effects” of mind–matter interaction [70]. The aim of the concept of irruption is to operationalize and then quantify the increased material underdetermination that is associated with the increased involvement of motivations. This is intended to be an intuitively intelligible concept; if we accept the possibility of a motivation-involving account of motivated activity, yet we also accept that only non-motivational processes constituting a behavior are measurable, then it necessarily follows that effects attributable to the involvement of motivations cannot directly show up as being motivated in measurements. Rather, by necessity, these effects can only be measured indirectly as a special subset of material changes that are *not* attributable to the activity’s material basis. Irruptions can, therefore, be approximated by the extent to which there is an increase in how unpredictable an activity is based on its material basis alone. In practice, this could, for example, involve quantifying how surprising recordings from the brain and body are with respect to past recordings.

This introductory sketch of irruption theory raises three questions that need to be addressed in more detail:(1)How can an irruption be quantified?(2)How can an irruption make a difference to behavior?(3)How can an irruption lead to appropriate behavior?

In response to these questions, irruption theory proposes three theses, namely, the theses of *irruption, scalability*, and *attunement*.

Irruption Thesis: The living body is organized as an *incomplete system* such that it is open to involvement of motivations via increased material underdetermination.

Scalability Thesis: The living body is organized as a *poised system* such that it amplifies microscopic irruptions to macroscopic fluctuations that can impact behavior.

Attunement Thesis: The living body is organized as an *attuned system* such that it responds to scaled up irruptions in a context-sensitive and adaptive manner.

Each of the three theses draws on existing research programs in embodied and enactive cognitive science and develops them in the context of irruption theory.

#### 5.2.1. Irruption Thesis

The primary ambition of irruption theory is to open the conceptual space required for a motivation-involving account of motivated activity. This ambition entails that the difference made by this motivational involvement would, in principle, preclude a complete description of the behavior in purely nonmotivational, i.e., material terms. In other words, to the extent that motivational involvement makes a difference to motivated activity, this behavior must correspondingly remain *underdetermined* by its current and past material states, including that of the whole body and, more generally, of the universe.

As posited by Axiom 3, there is a sense that all material processes can be considered to be underdetermined, but living systems seem to go further by being geared toward novelty generation, which is currently a topic of intense investigation [71,72,73]. Given the role that underdetermination plays in irruption theory, this would be a fitting context in which to revisit debates in the enactive approach on whether living systems are characterized by incompleteness due to the self-reference inherent in their metabolic self-production and operational closures [74,75]. If so, this would suggest that irruptions may need to be conceptualized as system-level changes rather than local changes. This is a topic for future work.

A practical upshot of irruption theory is that motivated activity is inherently unpredictable based on its material basis, and this suggests that irruptions can be indirectly quantified in terms their unpredictability. For this purpose, irruption theory can draw on information-theoretic measures of entropy—intuitively, a measure of the uncertainty that certain states will be observed—for neural, physiological, and behavioral signals. In general, the irruption thesis, thus, provides a novel interpretation for the growing evidence that information-theoretic entropy of brain activity is associated with all kinds of motivated activity [76].

For example, if we make the assumption that a person’s level of consciousness sets a general lower bound on their level of motivational involvement, there is already compelling evidence; levels of consciousness are associated with levels of neural entropy, a relationship which has led to the notion of the “entropic brain” [77,78]. Specifically, levels of consciousness that are associated with reduced awareness exhibit less neural entropy, such as dreamless sleep [79], while levels of consciousness that are deemed to be above normal waking consciousness exhibit increased neural entropy, especially the psychedelic state [80].

The specific contents of conscious experience have also been associated with the uncertainty of neural activity. For instance, increased variability of brain dynamics in a near-threshold auditory classification has been proposed as a neural signature of consciousness [81], which makes sense given that the perception of audible yet unclear stimuli is expected to elicit more motivational involvement to complete the task and, hence, more irruptions. The same reasoning applies to the finding that an increase in degrees of freedom of a perceptual task, which increases cognitive load to track multiple features, corresponds to an increase in neural entropy [82]. What these examples nicely highlight is that, while irruptions are unpredictable from the perspective of measurements of neurophysiological activity, they are not unpredictable *per se*, as they will tend to correlate with increased agential and subjective activity that, at least in the case of adult humans, can be indirectly accessed with first-person and second-person methods [83].

Irruption theory complements criticisms by Schurger et al. [84] of the so-called “readiness potential” that statistically precedes self-initiated movement, which has often been interpreted as demonstrating a lack of free will. Schurger et al. cast doubt on the causal relevance of this statistical pattern, and instead observed that the moment of movement onset is associated with increased stochastic fluctuations in neural activity. They highlighted a new theoretical challenge for attributing the movement to the participant; the source of increased fluctuations in neural activity that trigger the movement must be part of, or attributable to, the subject such that it counts as self-initiated. The irruption thesis provides a fitting response to this challenge because it precisely links the subject’s motivational involvement in an action to the relative underdetermination of neurophysiological activity.

Future work should look more closely at how irruptions relate to thermodynamic entropy. As a starting point, these information-theoretic measures of neural entropy arguably set a lower bound on the brain’s thermodynamic entropy [76]. However, a more direct link between irruptions and thermodynamic entropy could be developed, for example, in terms of a motivation-involving account of biological regulation. Essentially, this account would agree with ecological psychology’s proposal that cognitive systems form part of the larger class of far-from-equilibrium systems that are in accordance with the principle of maximum entropy production, yet which have the added flexibility of occasionally operating in a way that is thermodynamically arbitrary [24,36]. The key argument to develop would be to link this added thermodynamic flexibility to irruptions associated with motivated activity. This could also be developed into a novel perspective on life’s tendency of evolving toward forms of life with increased entropy production [23].

#### 5.2.2. Scalability Thesis

In general, the proportion of underdetermination of material processes that is attributable to irruptions would have to be exceptionally small-scale; otherwise, their consequences at the material level would have presumably already been noticed. Having said this, the theory of the thermodynamics of living systems still harbors many unknowns, and we may also be in for some big surprises as the measurement of the energetics of living tissue advances. If irruptions are most likely occurring at the smallest scales, then we need to posit the existence of mechanisms operating in the body that prevent their underdetermining effects from being washed out by large-scale material factors. We refer to this as the scalability thesis.

As the philosopher Jonas [85] realized, we can even account for scalability within the domain of classical physics. Consider the idealized situation of a macroscopic object arranged in an unstable equilibrium point, such as an upside-down cone perfectly balanced on its tip; even the smallest perturbation would make it fall over, with the consequence that the macroscopic change would in practice be entirely unpredictable. Nevertheless, such a mathematically fine-tuned situation is unlikely to be achieved in any real material system, and Jonas was ultimately more interested in considering the role of quantum physics in underdetermination. As the field of quantum biology starts to mature, this could be an interesting target for future work on irruption theory.

A macroscopic property that is more immediately relevant for biology is chaos; it refers to the unpredictability of long-term macroscopic trajectories of a system because it is sensitive to microscopic, nearly infinitely small, differences in initial conditions [86]. A related property of biological systems is so-called pink noise or 1/f noise, which is a form of scale-free dynamics, such that even the smallest perturbation can occasionally have macroscopic consequences [87]. The activity of the nervous system also exhibits 1/f noise, which dynamically changes with age, task demands, and cognitive states [88,89]. A fitting account of the origins of such dynamics in biological systems is *self-organized criticality*, whereby a system organizes itself so as to be poised to respond to perturbations in a scale-free manner [90]. In addition, in the context of consciousness science, there is a possibility to build on global neuronal workspace theory [91], especially its key concept of “ignition” of widespread neural activity [92].

These examples serve to illustrate that an irruption could occur at a microscopic scale, such that it would be remain practically unobservable, yet could still have consequences for the macroscopic scale of brain–body–environment interaction. Future work will need to work out a more detailed neurophysiological account of this scalability.

#### 5.2.3. Attunement Thesis

Irruptions cannot directly control a motivated activity’s material basis; they can only serve to counteract existing material constraints by increasing their underdetermination. Essentially, the changes induced by irruptions are arbitrary with respect to those material constraints, although it is important to consider that they happen in the context of a concrete agent–environment interaction process. Scaling irruptions up to a level that is relevant for behavior does not change this lack of directionality; they can loosen the organization of the current behavior but cannot force the formation of the behavior that is to come. Nevertheless, bodily activity spontaneously tends to self-organize in accordance with motivations. Irruption theory responds to this challenge with the thesis of attunement.

As the most minimal proof of concept of attunement, consider Ashby’s [69] classic nonrepresentational account of adaptive behavior in terms of instability-dependent arbitrary changes—“breaks”—in the living system’s organization, resulting in a stochastic search for an alternative stable configuration. Today, more sophisticated solutions can be found in work on embodied cognition, such as meta-stable attunement [93], implicit body memory at the individual and collective levels [94,95], and habits [96]. This work reveals that anticipatory action does not require a forward-looking central controller, nor does it necessarily depend on neural representations of future possibilities. Rather, our embodiment is shaped by its history of interactions to such an extent that the body is capable of appropriate movements for most situations. Work on “ultrafast cognition” demonstrates that not even feedback loops are needed for meaningful responses [97]. One possibility is to cash out more sophisticated forms of attunement, such as a tendency toward optimal grip on multiple relevant affordances, using a suitably adjusted version of the free energy principle [98]. The status of the Free energy principle with respect to the kind of enactive approach that has inspired much of irruption theory remains contentious [13]. However, future work on irruption theory could draw on some of its formalisms [99], e.g., to capture how the arbitrary openings in state space created by irruptions are closed off again with relevant activity.

In addition, irruption theory can draw inspiration from a tradition in agent-based simulation models, which illustrate how the agent’s capacity to temporarily neutralize constraints on the body’s processes can be beneficial for adaptive behavior. For example, it could facilitate the transition from too restrictive coping to a more open susceptibility to alternative task demands [42], and even just a neutralization of the influence of sensory or motor areas can facilitate action switching [100]. Intelligent action does not depend on a central controller but can emerge from a network of habits, which enables appropriate behavior to be solicited by the situation [96]. Once our bodies have become appropriately attuned, by evolution, development, and/or learning, the unfolding of future behavior can then be largely offloaded into the affordances and constraints of the agent–environment system. For example, the way in which an agent is embodied in the environment can condition the relative stability of its interaction patterns; a change in body morphology can spontaneously lead to a transition to the corresponding sensorimotor interaction pattern [101].

An even more compelling development comes from work on artificial neural networks with unsupervised associative learning, such as Hebbian learning (i.e., “neurons that fire together wire together”). When all the network’s neural states are occasionally “reset” to an arbitrary configuration—approximating a maximally scaled-up irruption pushing the system into far-from-equilibrium—and then allowed to converge to equilibrium again, the network will undergo a process of generalization over the set of visited equilibria [102]. It is a process of *self-optimization* of the attractor landscape such that the system’s chances of converging into equilibria that better coordinate the constraints imposed by the original state space increase over time (see Figure 4).

The conditions for this process of self-optimization are surprisingly simple, such that it can be replicated in various network architectures [103,104,105]. The self-optimization model can also be combined with self-organized criticality such that it has an intrinsic mechanism for the “reset” of neurons to arbitrary states across scales [106]. Given the potential generality of the self-optimization model—essentially, a system that is iterating between equilibrium and far-from-equilibrium dynamics plus some historicity akin to a principle of precedent—Froese and colleagues [11] proposed this to be a fitting model of the notion of adaptivity described by the enactive conception of life. In abstract thermodynamic terms, the living system’s need for openness to the environment’s energy flows corresponds to a far-from-equilibrium mode of operation (open system), while the system’s need for closedness for its structural integrity corresponds to an equilibrium mode (closed system). This interpretation is indicated by the similar visual arrangement of Figure 3 and Figure 4.

Froese and colleagues [11] also proposed that, given the key role assigned to the primordial tension in driving adaptivity, it is likely that more complex forms of life evolved other, more internalized ways of bringing about the conditions of self-optimization, including a process of generating the required itineration between these two distinct modes of operation. This is precisely what irruption theory can provide; given that adaptivity is a form of motivated activity, it can be the motivations, as such, in their expression as irruptions into the material basis of behavior, that could serve the role of the “resets” that push the system into the far-from-equilibrium mode of operation of an open system. In this way, irruptions could play the role of facilitating behavior switching, as well as contributing to overall long-term system–environment attunement. Irruption theory in combination with the self-optimization model, therefore, holds potential to provide a novel systems theoretic answer to the main concern raised against libertarian interpretations of motivated activity, i.e., regarding how an agent’s behavior can be effective even when it is underdetermined.

## 6. Concluding Remarks

Irruption theory has the ingredients required to develop a scientifically workable motivation-involving account of motivated activity in the context of a relaxed naturalism. It goes beyond current libertarian developments of the enactive approach to cognitive science with a key conceptual and methodological innovation; instead of staying with the mere logical possibility that an agent’s motivations, as such, can make a difference to its embodied activity because its material basis is not deterministic, the concept of irruption aims to capture the relative increase in material underdetermination of behavior resulting from an agent’s motivational involvement. An agent’s materiality and motivation both make a difference to its behavior, each in their own domain-specific way, without collapsing into an identity relation, which leads us to predict the presence of mind–matter cross-border effects.

Irruption theory does not propose to provide a mechanism of precisely how the motivations of an agent’s activity can increase the underdetermination of its material basis, which may ultimately turn out to be conceptually impossible even, in principle, due to the distinctive specificities of the two domains of phenomena that need to be brought together. Rather, the theory aims to advance the enactive approach, and cognitive science in general, by offering a novel way of conceptualizing, as well as quantifying, such mind–matter cross-border effects. The working hypothesis is that motivational involvement is correlated with irruptions, which are measurable as bursts of unpredictability of neurophysiological processes.

In summary, irruption theory provides a novel perspective on the accumulating evidence that motivated activities tend to be associated with increased information-theoretic entropy of neurophysiological processes. Essentially, it interprets such increased levels of entropy in terms of the existence of degrees of freedom that derive from agent-level motivations, and that, therefore, cannot be directly observed via neurophysiological measurement. It is likely that the metaphysical implications of this kind of relaxed naturalism, in which an agent’s motivations are taken to constitute a distinctive and intrinsic part of nature in addition to its materiality, will be difficult to accept for many. Others may find in this novel conceptualization a stimulating departure from the usual debates. Be that as it may, the main advantage of the current proposal is that it has methodological implications that can be developed into testable predictions, thus making it possible to more directly link some of the most exciting advances in the philosophy of mind with new experimental directions in cognitive science.

## Figures and Tables

**Figure 1 entropy-25-00748-f001:**
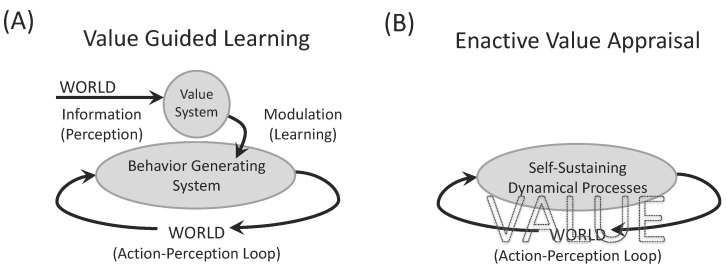
Distinct conceptualizations of the role of value in an organism’s behavior. (**A**) There is a popular class of cognitive architectures that feature a local mechanism that appraises values, such as a value system in the brain that generates reinforcement signals based on environmental states to guide the generation of behavior. (**B**) The enactive approach takes a holistic perspective on value as an aspect of all sense-making, which in turn is essentially an organism’s evaluation of the consequences of its actions in the world for the conservation of an identity. Figure redrawn from Di Paolo, Rohde, and De Jaegher [8].

**Figure 2 entropy-25-00748-f002:**
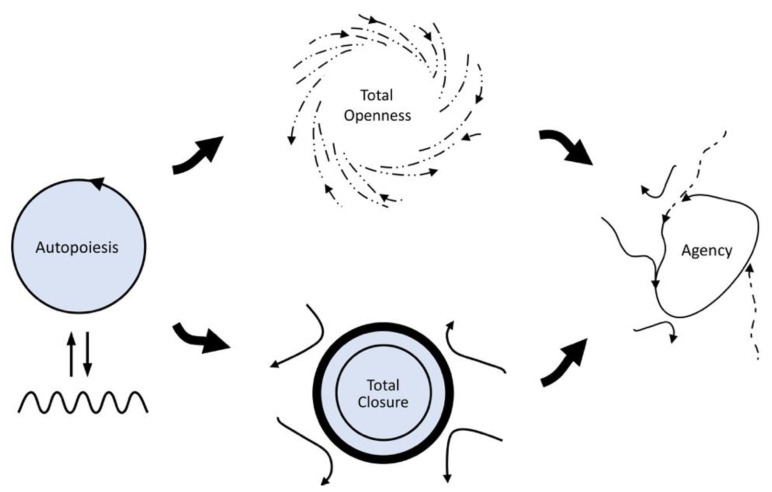
**The enactive approach to agency.** Starting from an autopoietic conception of life (left-hand side), the enactive approach highlights the intrinsic precariousness associated with life’s thermodynamic embodiment. There is a primordial tension between life’s need to be open to energy flows coming from the environment (middle top), and its need to maintain a structural identity that is distinct from the environment (middle bottom). Maximizing either of these needs exclusively would lead to death (dissolution and starvation, respectively). Thus, life requires agency; it must actively coordinate the partial satisfaction of one and the other of its needs over time. Figure taken from Froese and colleagues [11], redrawn from a figure by Di Paolo and colleagues [9].

**Figure 3 entropy-25-00748-f003:**
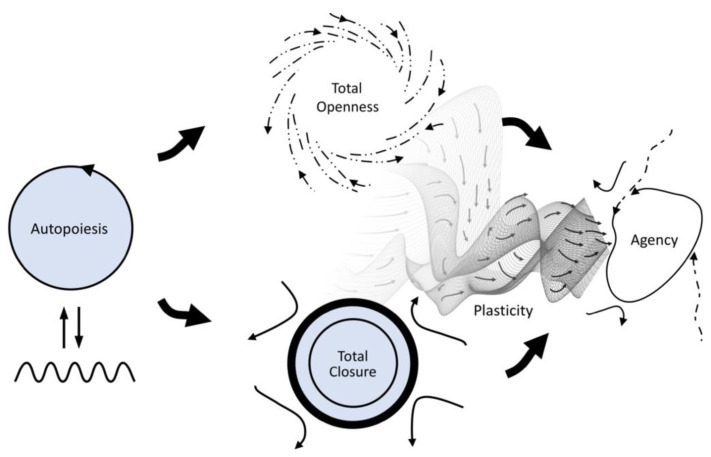
Updated schematic of the primordial tension of self-individuation, highlighting the role of path dependence and plasticity. See the caption of Figure 2 for details. On the right, it is now made explicit that the resolution of this tension requires a historical dimension. Figure taken from Froese, Weber, Shpurov, and Ikegami [11], redrawn and modified from Di Paolo, Buhrmann, and Barandiaran [9] (p. 135).

**Figure 4 entropy-25-00748-f004:**
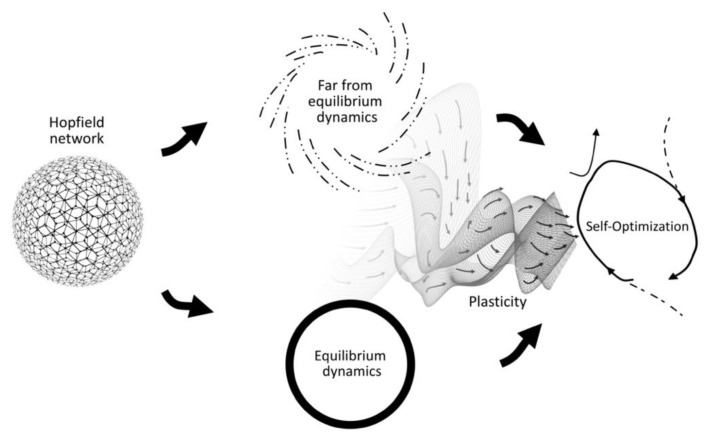
**Schematic of the self-optimization (SO) model.** On the left, there is the Hopfield network whose weighted connections represent the constraints in a coordination problem. The system tends to converge to an equilibrium, which is a possible solution to that problem, although, for complex problems this would typically only be a partial solution. In the middle, the two modes of the SO model are depicted: on the top, the network state is pushed into a far-from-equilibrium state, sufficient to escape the current basin of attraction; on the bottom, the network is allowed to converge into equilibrium. The system is switched between these two modes repeatedly. On the right, with the addition of unsupervised associative learning, the system begins a process of self-optimization: over time, the network re-organizes its relations such that it generalizes over past equilibria to reach otherwise unvisited, deeper equilibria. Figure copied from Froese, Weber, Shpurov, and Ikegami [11].

## Data Availability

Not applicable.

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
