# Peer review of "Irruption Theory: A Novel Conceptualization of the Enactive Account of Motivated Activity"

_entropy, 2023, doi:10.3390/e25050748_

Round 1
Reviewer 1 Report
This is a sophisticated, informed and informative discussion of some very basic concepts involving motivated behavior, consciousness, causality and what I would call material engagement. It covers a lot of ground. It presents a specific theory, Irruption Theory; explains the problems that it attempts to address, and the implications that would follow from that theory. The paper is clearly written (although it presumes an advanced knowledge of these issues) and well structured. I don’t have any major recommendations. I have a few minor comments.
- Froese suggests that all cognitive activity is a kind of motivated behavior. What about passive experiences – e.g., unbidden thoughts. Isn't this a cognitive activity unmotivated? Or is ‘activity’ strictly conceived of as a form of doing? So all cognitive activity is not equivalent to all cognitive processes, a cognitive activity (as a doing) must be motivated, but perhaps some cognitive processes do not have to be motivated [?].
- The distinction between a sense of agency and sense of ownership is a good one, but it might be helpful if the author could say a bit more about the sense of agency. It can be regarded as a complex phenomenon that operates prereflectively, or reflectively, and may involve social and normative factors. But the author later mentions some mechanism for a minimal form of agency. Is there a relation between sense of agency and a minimal agency, which seems to be attributed to an organism’s (non-conscious?) adapting to environment? Or any relation between the agency mechanism and the sense of agency? Also, this mechanism is said to allow for “the discrimination of system tendencies in terms of how threatening they are to the maintenance of its identity, and to compensate deleterious tendencies accordingly.” I wonder if Frederique de Vignemont’s idea of the “bodyguard hypothesis” – which relates sense of agency and sense of ownership might be relevant here. (de Vignemont, F., Pitron, V., & Alsmith, A. J. (2021). What is the body schema?. Body schema and body image: New directions, 3-17.). Likewise, in Figure 2, one might think that agency produces ownership (understood in terms of an ill-defined boundary).
- In Figure 3, does the plasticity/historicity imply that the identity/autonomy of the autopoietic organization changes over time, despite its effort to maintain itself?
- The notion of physical underdetermination is interesting. It seemingly creates a theoretical space for consciousness to have causal efficacy – since material processes underdetermine behavior. But does it explain how consciousness can actually have a causal effect on behavior? Also, it’s interesting to think of physical underdetermination in terms of the cognitivist ‘poverty of stimulation’ idea, which was challenged by Gibson. Does this make this version of enactivism closer to the cognitivist principle than to the ecological principle?
- Irruption would generate effects which “can only be measured indirectly as effects that are not attributable to its material processes.” I get this, but I wonder if putting it in this way sets up an opposition between agentive motivation versus material in a way that downplays the relational aspects – e.g., dynamical coupling. That is, if we attribute the effects to the agentive motivation, and not to the material aspects, do we get into a position of saying that it’s the agentive side that adds the value – versus, what I take to be an enactive principle that says it’s the relation or interaction or coupling between agent and environment that adds the value (or meaning, or whatever).
- It may be that I don’t understand the concept of arbitrariness used here (perhaps it has a technical definition), but on a relatively typical use of the term I’m not convinced that irruptions are arbitrary, as suggested in 5.2.3. At the level of bodily processes or of behavior, the underdetermination of the material doesn’t strike me as generating some kind of arbitrary process – I would think the underdetermination is in some sense specific in terms of what is left underdetermined, and is such that it allows the body or the agent a degree of freedom. If so, this would suggest a kind of selectivity (maybe creativity) rather than a kind of arbitrariness. Maybe this goes back to the previously mentioned down-playing of relationality. In terms of relationality, one could think of the underdetermination of the material as relative to the kind of organism involved, where different kinds of organisms might have different degrees of/or kinds of instabilities. One could think that the material environment could be, in some regard, more or less underdetermined for a frog than for a human, etc. So neither the underdeterminateness nor the irruption is arbitrary in any typical sense.
Author Response
I thank the reviewer for the encouraging and stimulating feedback. I have carefully gone through the manuscript in light of the comments in order to clarify the key points of the argument. I also detail my response to the specific comments below (in italics).
This is a sophisticated, informed and informative discussion of some very basic concepts involving motivated behavior, consciousness, causality and what I would call material engagement. It covers a lot of ground. It presents a specific theory, Irruption Theory; explains the problems that it attempts to address, and the implications that would follow from that theory. The paper is clearly written (although it presumes an advanced knowledge of these issues) and well structured. I don’t have any major recommendations. I have a few minor comments.
- Froese suggests that all cognitive activity is a kind of motivated behavior. What about passive experiences – e.g., unbidden thoughts. Isn't this a cognitive activity unmotivated? Or is ‘activity’ strictly conceived of as a form of doing? So all cognitive activity is not equivalent to all cognitive processes, a cognitive activity (as a doing) must be motivated, but perhaps some cognitive processes do not have to be motivated [?].
This is an interesting question. I now make it more explicit that the account is also supposed to include biological regulation, which is a kind of passive process of the body. But the example suggested by the reviewer is also interesting: what does this account have to say about experienced mental activity that is apparently not motivated? In future work it may be possible to unpack these experiences as irruptions coming from the side of the body into the side of the mind.
- The distinction between a sense of agency and sense of ownership is a good one, but it might be helpful if the author could say a bit more about the sense of agency. It can be regarded as a complex phenomenon that operates prereflectively, or reflectively, and may involve social and normative factors. But the author later mentions some mechanism for a minimal form of agency. Is there a relation between sense of agency and a minimal agency, which seems to be attributed to an organism’s (non-conscious?) adapting to environment? Or any relation between the agency mechanism and the sense of agency? Also, this mechanism is said to allow for “the discrimination of system tendencies in terms of how threatening they are to the maintenance of its identity, and to compensate deleterious tendencies accordingly.” I wonder if Frederique de Vignemont’s idea of the “bodyguard hypothesis” – which relates sense of agency and sense of ownership might be relevant here. (de Vignemont, F., Pitron, V., & Alsmith, A. J. (2021). What is the body schema?. Body schema and body image: New directions, 3-17.). Likewise, in Figure 2, one might think that agency produces ownership (understood in terms of an ill-defined boundary).
Again, these are are good questions! Right now I do not have all the answers worked out, but I am involved in a project that will systematically analyze the conditions for the sense of agency, and my hope is that irruption theory will provide a new perspective. Given that my claim is that agency makes a difference, but only indirectly so, this implies that the sense of agency cannot be based on control but perhaps instead on something like congruency of movement with motivation. This is a topic for future work, but I thank you for making me consider it.
- In Figure 3, does the plasticity/historicity imply that the identity/autonomy of the autopoietic organization changes over time, despite its effort to maintain itself?
Another excellent question! Yes, in future work it would be nice to work out an updated theory of autopoiesis based on the implications of irruption theory. My intuition is that the traditional focus on self-production will have to be replaced with a focus on self-differentiation, similar to other biological accounts of individuation.
- The notion of physical underdetermination is interesting. It seemingly creates a theoretical space for consciousness to have causal efficacy – since material processes underdetermine behavior. But does it explain how consciousness can actually have a causal effect on behavior? Also, it’s interesting to think of physical underdetermination in terms of the cognitivist ‘poverty of stimulation’ idea, which was challenged by Gibson. Does this make this version of enactivism closer to the cognitivist principle than to the ecological principle?
Another perceptive question! Indeed, I have not given an account of precisely how motivations can make a difference to an activity's material basis. I believe that this will not be possible in principle, because it requires crossing over between two domains of phenomena with distinctive characteristics that cannot be reduced to each other. I now clarified this point in the conclusion.
- Irruption would generate effects which “can only be measured indirectly as effects that are not attributable to its material processes.” I get this, but I wonder if putting it in this way sets up an opposition between agentive motivation versus material in a way that downplays the relational aspects – e.g., dynamical coupling. That is, if we attribute the effects to the agentive motivation, and not to the material aspects, do we get into a position of saying that it’s the agentive side that adds the value – versus, what I take to be an enactive principle that says it’s the relation or interaction or coupling between agent and environment that adds the value (or meaning, or whatever).
Thank you for highlighting this imprecision of my exposition. Even though my starting point is the enactive approach to value, I did not intend to make any commitment to where an agent's motivations originate, because the arguments focus on downstream consequences and are hence more general than that. No matter where the motivations come from, irruption theory could account for why they matter. I have clarified this point during my exposition of the enactive approach.
- It may be that I don’t understand the concept of arbitrariness used here (perhaps it has a technical definition), but on a relatively typical use of the term I’m not convinced that irruptions are arbitrary, as suggested in 5.2.3. At the level of bodily processes or of behavior, the underdetermination of the material doesn’t strike me as generating some kind of arbitrary process – I would think the underdetermination is in some sense specific in terms of what is left underdetermined, and is such that it allows the body or the agent a degree of freedom. If so, this would suggest a kind of selectivity (maybe creativity) rather than a kind of arbitrariness. Maybe this goes back to the previously mentioned down-playing of relationality. In terms of relationality, one could think of the underdetermination of the material as relative to the kind of organism involved, where different kinds of organisms might have different degrees of/or kinds of instabilities. One could think that the material environment could be, in some regard, more or less underdetermined for a frog than for a human, etc. So neither the underdeterminateness nor the irruption is arbitrary in any typical sense.
Good points again! In recent discussions in our research group we also had started considering that it would be important to highlight that not "anything goes", but that irruptions always happen within a concrete context of a history of agent-environment interactions that will shape how the space of possibilities will be closed toward a specific activity. I have tried to better contextualize irruptions throughout the manuscript. Thank you for pointing this out!
Reviewer 2 Report
This paper aims to introduce a philosophical position that would have implications for the scientific study of mind. The core of the paper is an argument given in section 5.1, that if we try to accept the premises ("axiom 1") that from a first-person perspective, our motivations make a difference to our behaviour, and ("axiom 2") that from a third-person perspective we cannot measure motivations and hence can't measure their causal effects, then we are forced to accept the conclusion ("axiom 3") that our behaviour must be "underdetermined" by our physical state.
I didn't find myself convinced by this, but it's not an unreasonable position to take. I didn't see anything fundamentally wrong with the arguments presented, and the paper does a good job of situating its content within an ongoing scientific discource, specifically the development of enavtive cognitive science. As such I don't see any strong barrier to publication.
However, there is one significant weakness in the paper that could be addressed, which is a certain amount of vagueness about what an irruption is really meant to be. The argument in section 5.1 reads as an argument for a classical Cartesian dualism, in which mental phenomena (our motivations) have a separate existence from the physical world, yet are still able to exert an influence on our behaviour. If one were to accept such a view one would then be forced to accept that behaviour is not completely determined by physical state, since otherwise there would not be any room for our (non-physical) motivations to exert an effect. The arguments of section 5.1 make sense in that context, even if I don't agree with them.
Yet much of the rest of the paper reads as if irruptions are merely random events, not caused by anything, mental or otherwise. We see this in particular in section 5.2.1, where a correlation between consciousness and information-theoretic entropy is given as evidence for irruption theory. (If irruptions were caused by our motivations then we wouldn't expect them to be random, it seems to me, since they would have to be correlated with the motivations that cause them.) We also see it strongly in section 5.2.2, which talks about phenomena like chaos, 1/f noise and self-organised criticality, which are associated with randomness or apparent randomness.
After reading the paper I am genuinely not sure which of these readings is the correct one, and feel it would be important to disambiguate. If the randomness reading is the correct one it might be a good idea to revise section 5.1, both to make sure it isn't misread as arguing for dualism and also to try and make it clearer how randomness per se can reconcile axioms 1 and 2.
On the other hand, if the dualist reading is the correct one then the paper seems to have a lot in common with the views of Roger Penrose, as set out in his books "the Emperor's new Mind" and "Shadows of the Mind" and in his work with Stuart Hameroff. That work proposes a specific way in which the mental could influence the physical via microscopic events at the quantum scale, and as such seems quite similar to some of the ideas in this paper. Regardless of whether this is the correct reading, it would be good to comment on the relationship to this work. (Especially since that work has been subject to some rather heavy criticism, and if the current work is successful it's likely that a lot of the same criticisms will be leveled at it.)
That concludes the main part of my review. The points below are more minor and are aimed mostly at the author.
The word "irruption" as a noun is never defined, and it would be useful to do so.
Why "irruption"? I had to look it up, but apparently the word means a sudden or violent entry or invasion or "bursting in". Perhaps the idea is that non-physical motivations violently "break in" to the physical world? But that seems an odd metaphor. (The word also has a separate technical meaning in ecology, but I'm sure that's not what's meant here.)
The paper often talks about the enative approach as if it's a single entity with goals of its own:
- "As part of its overall ambition [...] the enactive approach has been developing [...]" (lines 80-82);
- "it is possible to get the misleading impression that the enactive conception of life is proposing a theory of living systems focused on their stability" (lines 152-154);
- "the enactive approach has opted to make conceptual room for an emergent role of normativity at the system level." (lines 180-181);
- "the enactive approach evidently does not want to return to either vitalism or cognitivism [...]" (lines 247-248)
- "The enactive approach has not pursued either of these possible responses, but not for a lack of opportunities" (lines 265-266)
- "Why does the enactive approach resist either of these directions even if it means leaving both sets of criticisms unanswered?" (lines 268-271)
- "the enactive approach’s unwillingness or incapacity to respond to these critics" (lines 275-276)
- references to the enavtive approach's "ambitions" on lines 290, 297
- reference to the enative approach not being "tempted" on line 301
- the enactive approach being initially "keen" on simulation methods on line 312
- the enactive approach "explicitly [distancing]" itself from autopoiesis on line 328
Many of these are grammatically jarring, but aside from that I guess I object to it partly because if you include people like O'Reagan and Noe, or even Hutto and Myin, then enactivism is a fairly broad church and these comments clearly don't apply to all of it. In general it seems odd to refer to an academic school of thought as if it had intentions of its own. I have the impression that most of these comments are actually referring to the views of a specific small group of authors (perhaps even just one author), and I think it would be more scholarly to attribute such intentions to specific individual authors by name, rather than to "the enactive approach" as a whole.
The argument on lines 196-201 was too condensed for me to understand - this part could be expanded.
Around lines 203-217 the paper talks about a lack of engagement between the enactive approach and thermodynamics. However, there have been several approaches that tried to relate these, such as the work of Alvaro Moreno, Kepa Ruiz-Mirazo, among several others. Although it is somewhat to the side of the main point, it would be worth adding some citations here, since there clearly has been work that addresses this point.
I didn't understand what was meant on lines 233-237, "It does not help that the notion of sense-making, which had motivated the elaboration of autopoietic theory with this account of adaptivity, was in turn motivated by appealing to the richly graded normativity of our lived experience. An appeal to characteristics of the first-person perspective does not [make?] scientific concepts more workable, at least not without further theoretical work." What doesn't it help with? A citation seems to be missing for the first sentence, perhaps to Di Paolo's paper 'Autopoiesis, Adaptivity, Teleology, Agency'. Is the second sentence intended as a statement of opinion or something else?
"Libertarian" seemed an odd choice of word, being associated with a political movement that has its own, rather different, philosophical position - if this isn't an established term I would suggest using a different word.
I didn't understand how the work cited in section 5.2.3 could be relevant to the paper's main thesis. Most of the work is to do with computer models, and computers are not underdetermined almost by definition. Maybe there is some way in which this section is relevant anyway, but it wasn't clear to me.
Author Response
I appreciate that the reviewer took the time to provide detailed feedback on this manuscript. I have taken all the comments into consideration while carefully revising the manuscript to improve clarity of exposition. I provide point-by-point responses to the comments below (in italics).
This paper aims to introduce a philosophical position that would have implications for the scientific study of mind. The core of the paper is an argument given in section 5.1, that if we try to accept the premises ("axiom 1") that from a first-person perspective, our motivations make a difference to our behaviour, and ("axiom 2") that from a third-person perspective we cannot measure motivations and hence can't measure their causal effects, then we are forced to accept the conclusion ("axiom 3") that our behaviour must be "underdetermined" by our physical state.
I didn't find myself convinced by this, but it's not an unreasonable position to take. I didn't see anything fundamentally wrong with the arguments presented, and the paper does a good job of situating its content within an ongoing scientific discource, specifically the development of enavtive cognitive science. As such I don't see any strong barrier to publication.
However, there is one significant weakness in the paper that could be addressed, which is a certain amount of vagueness about what an irruption is really meant to be. The argument in section 5.1 reads as an argument for a classical Cartesian dualism, in which mental phenomena (our motivations) have a separate existence from the physical world, yet are still able to exert an influence on our behaviour. If one were to accept such a view one would then be forced to accept that behaviour is not completely determined by physical state, since otherwise there would not be any room for our (non-physical) motivations to exert an effect. The arguments of section 5.1 make sense in that context, even if I don't agree with them.
The reviewer is correct that the proposal rejects the identification of motivations with materiality as it is described by the physical sciences, even if all motivated behavior also has a material basis. This is because an agent's motivations have associated conditions of normativity that their material basis alone does not have. However, dualism is not the only interpretation of this rejection of identity. Another possibility, which is the one briefly suggested in the article, is to adopt a relaxed naturalism according to which both motivations and materiality form part of reality but not in a way that would allow their full separability.
Yet much of the rest of the paper reads as if irruptions are merely random events, not caused by anything, mental or otherwise. We see this in particular in section 5.2.1, where a correlation between consciousness and information-theoretic entropy is given as evidence for irruption theory. (If irruptions were caused by our motivations then we wouldn't expect them to be random, it seems to me, since they would have to be correlated with the motivations that cause them.) We also see it strongly in section 5.2.2, which talks about phenomena like chaos, 1/f noise and self-organised criticality, which are associated with randomness or apparent randomness.
I thank the reviewer for bringing this ambiguity to my attention. I hope that by carefully revising the paper throughout, I have now made it clear that the arbitrariness of irruptions is with respect to past measurements of the material basis of motivated activity. In other words, irruptions appear as random if only predicted based on measurements of the system's past material states. But of course the irruptions correlate with moments of elevated agential involvement of motivations, and so they are not arbitrary from this more inclusive perspective.
After reading the paper I am genuinely not sure which of these readings is the correct one, and feel it would be important to disambiguate. If the randomness reading is the correct one it might be a good idea to revise section 5.1, both to make sure it isn't misread as arguing for dualism and also to try and make it clearer how randomness per se can reconcile axioms 1 and 2.
Yes and no. The point of irruption theory is precisely to question the necessity of binary black-or-white thinking. Irruptions are neither random nor an appeal to dualism. They are cross-border effects that result from motivations and materiality being not one and not two.
On the other hand, if the dualist reading is the correct one then the paper seems to have a lot in common with the views of Roger Penrose, as set out in his books "the Emperor's new Mind" and "Shadows of the Mind" and in his work with Stuart Hameroff. That work proposes a specific way in which the mental could influence the physical via microscopic events at the quantum scale, and as such seems quite similar to some of the ideas in this paper. Regardless of whether this is the correct reading, it would be good to comment on the relationship to this work. (Especially since that work has been subject to some rather heavy criticism, and if the current work is successful it's likely that a lot of the same criticisms will be leveled at it.)
I thank the reviewer for making this association with Penrose's work on consciousness, and for highlighting the dangers that are waiting in that direction. In the revised version I make a slightly more explicit reference to quantum biology, but I feel that a proper engagement with that literature is beyond the scope of the current article.
That concludes the main part of my review. The points below are more minor and are aimed mostly at the author.
The word "irruption" as a noun is never defined, and it would be useful to do so.
Why "irruption"? I had to look it up, but apparently the word means a sudden or violent entry or invasion or "bursting in". Perhaps the idea is that non-physical motivations violently "break in" to the physical world? But that seems an odd metaphor. (The word also has a separate technical meaning in ecology, but I'm sure that's not what's meant here.)
I thank the reviewer for prompting me to include details about the reasons for choosing the word "irruption". When I first introduce the noun I now also mention these original Latin meanings and why they are relevant.
The paper often talks about the enative approach as if it's a single entity with goals of its own:
- "As part of its overall ambition [...] the enactive approach has been developing [...]" (lines 80-82);
- "it is possible to get the misleading impression that the enactive conception of life is proposing a theory of living systems focused on their stability" (lines 152-154);
- "the enactive approach has opted to make conceptual room for an emergent role of normativity at the system level." (lines 180-181);
- "the enactive approach evidently does not want to return to either vitalism or cognitivism [...]" (lines 247-248)
- "The enactive approach has not pursued either of these possible responses, but not for a lack of opportunities" (lines 265-266)
- "Why does the enactive approach resist either of these directions even if it means leaving both sets of criticisms unanswered?" (lines 268-271)
- "the enactive approach’s unwillingness or incapacity to respond to these critics" (lines 275-276)
- references to the enavtive approach's "ambitions" on lines 290, 297
- reference to the enative approach not being "tempted" on line 301
- the enactive approach being initially "keen" on simulation methods on line 312
- the enactive approach "explicitly [distancing]" itself from autopoiesis on line 328
Many of these are grammatically jarring, but aside from that I guess I object to it partly because if you include people like O'Reagan and Noe, or even Hutto and Myin, then enactivism is a fairly broad church and these comments clearly don't apply to all of it. In general it seems odd to refer to an academic school of thought as if it had intentions of its own. I have the impression that most of these comments are actually referring to the views of a specific small group of authors (perhaps even just one author), and I think it would be more scholarly to attribute such intentions to specific individual authors by name, rather than to "the enactive approach" as a whole.
The reviewer is correct that there are several strands of enactivism. In some respects what I write is relevant for all of them, but when I refer to the "enactive approach" I refer to one specific strand that is referred to by the other strands as "autopoetic enactivism" (see, e.g., footnote 1). There is quite a few authors working within the scope of the enactive approach, so it makes sense to refer to the approach rather than individuals.
The argument on lines 196-201 was too condensed for me to understand - this part could be expanded.
Thank you for highlighting this point. In fact, I have another whole paper under review which precisely provides a deeper and more systematic look at the relationship between the current proposal and the physics of life. However, it would go beyond the scope of the current paper to expand too much on this point.
Around lines 203-217 the paper talks about a lack of engagement between the enactive approach and thermodynamics. However, there have been several approaches that tried to relate these, such as the work of Alvaro Moreno, Kepa Ruiz-Mirazo, among several others. Although it is somewhat to the side of the main point, it would be worth adding some citations here, since there clearly has been work that addresses this point.
I have now added a citation to work by Ruiz-Mirazo and Moreno.
I didn't understand what was meant on lines 233-237, "It does not help that the notion of sense-making, which had motivated the elaboration of autopoietic theory with this account of adaptivity, was in turn motivated by appealing to the richly graded normativity of our lived experience. An appeal to characteristics of the first-person perspective does not [make?] scientific concepts more workable, at least not without further theoretical work." What doesn't it help with? A citation seems to be missing for the first sentence, perhaps to Di Paolo's paper 'Autopoiesis, Adaptivity, Teleology, Agency'. Is the second sentence intended as a statement of opinion or something else?
Yes, there was a typo in that sentence, which has now been corrected. I have also revised the paragraph to clarify the main point and I have added a citation to Di Paolo (2005).
"Libertarian" seemed an odd choice of word, being associated with a political movement that has its own, rather different, philosophical position - if this isn't an established term I would suggest using a different word.
Libertarianism is a well-known tradition in the philosophy of free will. I have now clarified this.
I didn't understand how the work cited in section 5.2.3 could be relevant to the paper's main thesis. Most of the work is to do with computer models, and computers are not underdetermined almost by definition. Maybe there is some way in which this section is relevant anyway, but it wasn't clear to me.
The computer models are introduced as thought experiments to show that stochastic dynamics can facilitate adaptive behavior. They are indeed only simulations and do not feature genuine indeterminacy. But the main point is to use them to look at the downstream consequences of stochastic dynamics, so in this case it does not matter whether that stochasticity is ultimately derived from a deterministic or nondeterministic source.